# Size-Mediated Sea Turtle Behavioral Responses at Artificial Habitats in the Northern Gulf of Mexico

**DOI:** 10.3390/ani13010114

**Published:** 2022-12-28

**Authors:** Tabitha Renee Siegfried, Jackson Reimer, Emma Roberto, Christopher Noren, Alex Vidal, Kristi Dixon, Morgan DuBois, Susan E. Piacenza

**Affiliations:** 1Department of Biology, University of West Florida, Pensacola, FL 32514, USA; 2Darling Marine Center, University of Maine, Walpole, ME 04573, USA; 3United States Fish and Wildlife Service, Maryland Fish and Wildlife Conservation Office, Annapolis, MD 21401, USA; 4Department of Fisheries, Wildlife and Conservation Sciences, Oregon State University, Corvallis, OR 97331, USA

**Keywords:** sea turtle wariness, artificial reefs, loggerhead sea turtles, Kemp’s ridley sea turtles, green sea turtles, stereo-video camera, size-mediated behaviors, northern Gulf of Mexico

## Abstract

**Simple Summary:**

In-water behavioral observation studies of sea turtles are lacking primarily because they are highly migratory throughout their lifecycles. However, utilizing new technology, such as remote stereo-video cameras, allows researchers to collect size-specific data and view relatively natural sea turtle behavior. In this study, we completed stereo-video camera surveys at local artificial reefs, piers, and jetties within the northern Gulf of Mexico from May 2019 to August 2021. We remotely measured sea turtle straight carapace length, assessed wariness using minimum approach distance (MAD) between the turtle and camera operator, and documented behaviors during encounters. Our results indicated that green sea turtles had the smallest mean MAD before a startle response occurred, but the largest range compared to loggerhead and Kemp’s ridley sea turtles. Using a linear mixed model, we found that the size of the turtle was an important predictor of MAD. Overall, our results suggest that larger sea turtles became startled at longer distances compared to the smaller individuals surveyed across all the species studied. This study supports the use of stereo-video camera systems as a non-invasive tool to conduct surveys to assess sea turtle behavioral observations in relation to body size.

**Abstract:**

Our understanding of size-specific sea turtle behavior has lagged due to methodological limitations. However, stereo-video cameras (SVC) are an in-water approach that can link body-size and allow for relatively undisturbed behavioral observations. In this study, we conducted SVC dive surveys at local artificial reefs, piers, and jetties in the northern Gulf of Mexico (nGOM) from May 2019 to August 2021. Using SVCs, we measured sea turtle straight carapace length, documented behaviors, and quantified wariness by assessing minimum approach distance (MAD). In green sea turtles (*Chelonia mydas*), the observed MAD ranged from 0.72 to 5.99 m (mean 2.10 m ± 1.10 standard deviation (SD), *n* = 73). For loggerhead sea turtles (*Caretta caretta*), the MAD ranged between 0.93 and 3.80 m (mean 2.12 m ± 0.99 SD, *n* = 16). Kemp’s ridley sea turtles (*Lepidochelys kempii*) were similar to loggerheads, and MAD ranged from 0.78 to 3.63 m (mean 2.35 m ± 0.99 SD, *n* = 8). We then evaluated what biological factors could impact the MAD observed by species, but we excluded Kemp’s ridleys as the sample size was small. Using a linear mixed model and model selection based on AICc, the top ranked model for both green and loggerhead sea turtles included SCL as the most important factor influencing MAD. MAD did not vary with habitat type for either species. Our results showed that larger individuals, regardless of species, have a greater wariness response, becoming startled at greater distances than smaller individuals. The findings of our study support the use of SVC as an accessible, non-invasive tool to conduct ecologically relevant in-water surveys of sea turtles to link behavioral observations to body size.

## 1. Introduction

The behavior of animal species shifts through its lifespan and is often correlated with body size [1]. For example, grey mouse lemur (*Microcebus murinus*) boldness scales with body size, where younger males are less bold than older and larger individuals, and they become bolder as they age [1]. In largemouth bass (*Micropterus salmoides*), consumption rate and foraging activity is influenced by body size, in the presence of a visible predator, while prey type is unchanged [2]. Marine iguanas (*Amblyrhynchus cristatus*) display differences in foraging duration and intake per bite that scale with body size, where smaller animals have higher foraging efficiency [3]. Behavior often changes as food selectivity (i.e., gape size) or predation and disease risks shift with size and can drive how animals interact with their environment as they perceive food availability and risk of any form [4,5,6,7,8,9]. However, in marine ecosystems, in situ observations of behavior in relation to body size have been obstructed due to methodological limitations. As such, our understanding of marine animal behavior, and in particular size-specific variation, has lagged in comparison to terrestrial ecosystems [10,11]. Nonetheless, emerging in-water methods, such as biotelemetry, now allow novel insights into animal behavior underwater.

Methods of biotelemetry to study animal behavior include satellite transponders, data loggers, remote operated vehicles, animal-borne cameras, and aerial surveys [12,13,14,15,16,17,18,19,20,21]. Until recently, none of these methods could specifically link behavioral observations to body size and interactions with the environment. However, stereo-video cameras (SVC) are a video-based in-water non-invasive approach that has vastly grown in acceptance as a cost-effective method for estimating animal body size [22,23,24,25,26,27]. SVC systems comprise two cameras that record overlapping perspectives to create a three-dimensional image allowing for length measurements to be estimated. SVCs measure body length and other morphometrics, but the video footage can also be used for behavioral observations. One potential advantage to using the SVC to discern marine animal behavior is the ability to observe individuals in their natural, relatively undisturbed habitat, as well as potentially assess an animal’s response to anthropogenic disturbances. Fortunately, advances in stereo-video photogrammetry make measuring in-water body length easier and more accurate compared to diver-based visual estimates or single camera systems [24,25,26,28]. This is advantageous for endangered species, such as sea turtles, which have until recently had limited behavioral studies in natural or artificial habitats (but see, e.g., [14,18,29,30,31,32,33,34]).

SVC surveys have been conducted for sea turtles in the northern Gulf of Mexico (nGOM), the Bahamas, and the Philippines [24,27,35,36]. SVCs allow researchers to achieve a more complete picture of a given sea turtle population and link behavior to body size while observing ecological interactions. Indeed, sea turtles are particularly challenging for in situ ecological studies in general, and behavioral analyses specifically [37]. Thus, applying stereo-video photogrammetry is beneficial on two fronts: augmenting size-based population studies and contributing to knowledge of their behavior.

Where densities are sufficiently high, sea turtles can have positive ecosystem-level impacts and contribute to resilience across nearly entire oceanic basins [38,39,40,41,42,43,44]. Sea turtles are found in temperate to tropical neritic and oceanic waters; however, despite intensive conservation efforts, sea turtles continue to be harvested in many parts of their ranges and are also bycaught in many fishing sectors [14,43,44,45,46,47]. Having a better understanding of their behavior may help mitigate fishing and other anthropogenic impacts that sea turtles face [14,48]. For example, within the nGOM, sea turtles have been found occupying local fishing piers, artificial reefs, pipelines, and man-made rock jetties where these artificial habitats may have different types of threats compared to natural habitats [35,36,49,50,51,52]. Population recovery of green *(Chelonia mydas*)*,* loggerhead *(Caretta caretta*)*,* and Kemp’s ridley (*Lepidochelys kempii*) sea turtles will potentially shift behavior and habitat use by way of the ideal free distribution as localized density-dependent effects becomes more prominent [53,54,55]. While a handful of in-water based studies on sea turtle behavior do exist, none directly link behavior to body size (e.g., [14,15,18,29,31,56]). So, then, how does sea turtle behavior shift with body size across species?

Given the differences in survival and predator risk, we predict that behavior would change with body size and age class. The aim of this study was to use SVCs to observe sea turtle behavior at nearshore artificial reefs, jetties, and fishing piers in the nGOM and evaluate differences across species and body size. Additionally, we assessed sea turtle behavioral response towards divers, quantifying behavioral shifts by focusing on boldness-wariness as one dimension of behavior by using minimum approach distance [57]. Here, we provide an approach to quantify different dimensions of sea turtle behavior using non-invasive in-water photogrammetry. The results from this study can be used to improve the understanding of sea turtle behavior, habitat use, and to improve conservation for threatened and endangered marine species.

## 2. Materials and Methods

We conducted shore-based SVC dive surveys (*n* = 64) at local Florida artificial reefs, piers, and jetties from Santa Rosa to South Walton Counties in the nGOM from May 2019 to August 2021, with dive sites selected opportunistically and visited on a rotating basis (Figure 1), attempting to visit all sites equally. No scientific research permits were required as the SVC surveys were non-invasive, and we followed individuals for ≤5 min (per guidance from NOAA Office of Protected Resources and Florida Fish and Wildlife Conservation Commission).

We inspected each site for sea turtles by swimming around all exposed hard substrate (i.e., reef modules, rocks, pier pilings, etc.) with the average survey lasting approximately 30 min, as part of long-term SVC monitoring project [35]. To ensure diver safety and to assist in collecting additional data (i.e., facial identification photos, water temperature, maximum depth, visibility, survey time, and weather conditions), two or more divers were present at every dive. For every turtle encountered during a survey, we recorded species, time of encounter, and presence of flipper tags. During this long-term monitoring project, we used the SVC to measure straight carapace length (SCL). However, after thoroughly inspecting all videos, unique behaviors were observed and the current study, presented here, was based on the archival video data.

Dive transect videos were initially reviewed to identify all unique sea turtle encounters within the survey to allow for easier behavioral analysis. Video reviewers assessed all footage and documented time of encounter, end time of encounter period, and relevant biological and behavioral data. We deemed the initial interaction as the earliest instance of visual sighting, we determined the end time of the encounter period when the sea turtle was lost visually and either not encountered again or had changed behavior upon re-encounter.

We used SeaGIS EventMeasure software v.5.22 (SeaGIS Pty Ltd., Bacchus Marsh, Victoria, Australia) to collect SCL by selecting the nuchal scute and the supracaudal scute in both the right and left camera frames [24,58]. We used standardized protocols to calibrate the cameras and collect SCL adapted for sea turtles (Harvey and Shortis 1996, Harvey and Shortis 1998, Siegfried et al., 2021).

We used minimum approach distance (MAD) to quantify one dimension of sea turtle behavior: wariness to divers [57]. MAD is defined as the shortest distance that the researcher was able to approach the sea turtle before the turtle exhibited a startle response (i.e., increase flipper stroke, bolting, change from resting to swimming, etc.). We also used SeaGIS EventMeasure software to measure MAD from the video footage, independent from the SCL measurement. First, we identified the video frame with the closest distance to between the SVC and the turtle. We then used a visible body part in the paired right and left video frames, such as the head, flipper, or carapace, to collect a measurement, but only extracted the range, which is the distance between the SVC and the object of interest. For many of the encounters, the viewpoint of the turtle at the closest distance varied; therefore, the part of the turtle that was measured to obtain the range was opportunistically selected depending on which part of the body was visible for each individual turtle. Notably, it is possible to reduce the length of time required to collect MAD values by selecting the same geospatial 3D point on the turtle in the left and right camera frames to automatically generate a range from the turtle to the diver in EventMeasure, without selecting a measurement vector, rather than conducting an arbitrary measurement as we did here [57].

We identified, categorized, and recorded all behaviors that were observed in the video files (Table 1; Appendix A). For consistency across encounters and to minimize variability in observations, we only used the left camera of the SVC. In addition, we recorded the total time of the encounter, time at startle, behavior prior to a startle response, and intensity of the startle for each sea turtle encounter. Additional biologically relevant data were recorded, such as dive method (i.e., snorkel or SCUBA), diver approach direction, water visibility, habitat type (i.e., artificial reef, pier, jetties). We also recorded data related to interactions with other organisms including any intraspecific behaviors (i.e., multiple turtles associated together or interacting with each other), food ingested, cleaner species observed, and commensal species (i.e., shark suckers, *Echeneis naucrates*).

We defined a startle response as a change in behavior because of the researcher’s presence [59]. We were initially interested in understanding differences in startle intensities across species. Therefore, we assigned a startle intensity value from 0 to 2 for each diver approach, where (0) indicated no startle response during the diver encounter, (1) indicated a perceptible change in behavior, including a slight change in direction of movement or activity, (but no strong change in direction or activity), and (2) indicated a strong, perceptible startle behavior, such as bolting (immediately swimming away with frequency and strong flipper strokes, and/or a 90–180 degree change in direction). The higher the startle intensity, the more abrupt the behavior change was. Ultimately, in the statistical analysis we changed the startle variable to a binary response (no startle or startle) as defining the severity of a startle was somewhat subjective. We defined a no startle response as an encounter in which the turtle displayed no direct change in behavior throughout the entire encounter. Startle time was defined as the earliest visual indication of behavior towards the presence of researchers for each encounter. We reviewed the entirety of the encounter, pre-startle to post-startle, while noting which behaviors occurred prior to a startle response, if it occurred, and after the startle response. We interpreted the behavior prior to a startle response as the natural behavior of the sea turtle. We assumed that all behaviors that followed were influenced by the diver. If no startle occurred, we assumed all behaviors observed to be the natural behavior of the turtle. Following video analysis, each video was assigned to one reviewer and one quality assurance-quality control (QA/QC) analyst to ensure consistency across all encounters and individual researchers. Thus, all turtle encounters were reviewed independently at least twice. If there was disagreement between the initial reviewer and QA/QC analyst, the pair then re-reviewed the video a third time to mutually agree on type or timing of behavioral observations. If necessary, a third analyst was then consulted to find consensus in behavioral observations.

To address how behavior varied across species, body size, and habitat type, we statistically evaluated potential correlates of MAD as a function of body size (i.e., SCL; cm), habitat type, and dive method in a log-transformed linear mixed effects model (LMM) for green and loggerhead sea turtles, separately. While we did observe Kemp’s ridley sea turtles, the sample size was too small for robust statistical analysis (*n* = 8). Multiple visits were made to each survey site (location); therefore, location was treated as a repeated effect. We evaluated model diagnostic tests of an untransformed LMM, such as residuals to fitted values, frequency distribution of residuals, and quantile-quantile plots, and the plots displayed heterogeneity in variance. Therefore, the response variable, MAD, and the explanatory variable, body size, were log-transformed. We evaluated factors influencing body size using the package *lme4* in RStudio v. 4.2.1 [60].

To better identify the explanatory variables that influenced MAD, we used the information-theoretic approach for model selection based on Akaike Information Criterion correction (AICc) for small sample size [61,62]. We used the dredge function in the R package *MuMin* [63]. Models with ΔAICc < 2 from the top-ranked model were retained in the confidence model set. When more than one model was retained in the confidence set, we examined relative importance (R.I.) of each variable included in the confidence model set. High values of R.I. results from variables occurring in large proportion of the confidence model set, indicating further support for the variables [61]. Lastly, we examined the 95% confidence intervals of all explanatory parameters in the confidence set to identify uninformative parameters, i.e., parameters that had confidence intervals crossing zero [61,64,65]. All candidate models were tested against our global model:(1)$$\ln\left(\mathrm{MAD}\right)=\beta_{0}+\beta_{1}\times \ln{\left(\mathrm{SCL}\right)}^{t}+\beta_{2}\times {\mathrm{Habitat}\;\mathrm{Type}}^{t}+{\;\beta }_{3}\times {\;\mathrm{Dive}\;\mathrm{Method}}^{t}+u_{i,j}^{t}\times \gamma_{i}u_{i,j}+\varepsilon_{i,j}$$
where β_0_ is the intercept, ln(SCL) is the ln-transformed body length, *u_i,j_* is the covariate vector of the j-th member of site *i* for random effects, *γ_i,j_* is the random effects parameter, and *ε_i,j_*~N(0, σ^2^)). All analyses were performed in R (R Development Core Team, 2021) and RStudio (R Studio, Inc., Boston, MA, USA).

## 3. Results

During our study, 107 unique sea turtle encounters were recorded across 16 dive sites. Of these, 89% (*n* = 65) of green sea turtles were classified as juveniles (SCL < 65 cm), 8.2% (*n* = 6) were considered subadults (65 < SCL < 90) and 2.7% (*n* = 2) were considered adults (SCL > 90 cm) [66]. The overall mean SCL for greens was 48.33 cm ± 16.44 cm standard deviation (SD), *n* = 73 (Figure 2). Of the loggerhead sea turtles observed, 62.5% (*n* = 10) were classified as subadults (SCL < 82 cm), and 37.5% (*n* = 6) were considered adults (SCL > 82 cm, with a mean SCL 79.02 cm ± 12.19 cm SD, *n* = 16) [66]. Of the Kemp’s ridley sea turtles, the mean SCL was 50.57 cm ± 6.85 cm SD (*n* = 8), and all were classified as subadults (20 < SCL < 50) [67,68].

Ten encounters did not allow for accurate SCL measurements and therefore were excluded from the statistical analysis. However, all 107 unique encounters were documented in terms of behavioral observations. Observed behaviors were then broken into two categories: pre-startle (Figure 3) and startle (Figure 4).

For pre-startle behaviors for green sea turtles, we most frequently observed swimming forward (*n* = 21) and no startle (*n* = 21; Figure 3). For loggerhead sea turtles, we most often observed resting under the reef modules (*n* = 7). We observed Kemp’s ridley sea turtles most often resting under hard structures, such as a reef module (*n* = 5). For both loggerhead sea turtles and green sea turtles, the primary behavior observed after being startled by the researcher was swimming forward (*n* = 52, green sea turtles; *n* = 11, loggerhead sea turtles). Notably, few individuals exhibited significant behavior changes (i.e., resting to bolting). Only eight encounters resulted in the sea turtle bolting, and 31 encounters resulted in no startle. For all species of sea turtles, a startle most frequently resulted in swimming, followed by increased flipper strokes, and changing direction (Figure 4).

We had some rarely recorded behavioral observations: self-cleaning, intraspecific interactions, and feeding. Green sea turtles were observed feeding on five separate encounters (all juveniles, size range from 24.5 to 59.7 cm SCL), but feeding was not observed in Kemp’s ridley or loggerhead encounters. However, both green (*n* = 3) and loggerhead sea turtles (*n* = 1) were observed cleaning; this included self-cleaning on reef modules or commensal cleaning with various fish species. On two occasions, sea turtles (one loggerhead, 79.4 cm SCL, and one green, 59.7 cm SCL) were observed using artificial reef modules as self-cleaning stations. While the loggerhead was self-cleaning on the artificial reef, fish also cleaned the loggerhead’s body. We observed sea turtles laying underneath the bottom disc of the reef module to rub their carapace along the reef module (Appendix A). We had several intraspecific interactions where two turtles were closely associated with each other. On two separate occasions, two small juvenile green sea turtles were resting on rubble at Destin Jetties (31.5 and 30.7 cm SCL, and 34.6 and 30.9 cm SCL, respectively). Additionally, two green sea turtles were observed competing for a PVC pipe partially buried underneath Navarre Beach Fishing Pier for cleaning; however, this video was not taken with the SVC, and therefore, was not included in our analysis (Appendix A). While this may have been the first recorded instance of size-specific behaviors observed, these rarely recorded behaviors were not included in the formal statistical analysis.

We used the minimum approach distance (MAD) to assess the closest possible distance a diver approached the sea turtle before a startle behavior was elicited. Green sea turtles (*n* = 73) had the largest range (0.72–5.99 m; Table 2) but the smallest mean MAD, while both loggerhead sea turtles and Kemp’s ridley sea turtle ranges were smaller but similar to each other, but the mean MADs were greater (Table 2).

We evaluated several biological factors (i.e., diver method, SCL, and habitat type) that could impact the MAD. Since green sea turtles and loggerhead sea turtles might respond differently, we separated the analysis of biological factors and MAD by species. For green sea turtles, the model confidence set included three models (Table 3). In the top ranked model, SCL and location (as the random effect) most influenced the MAD. While in the second model, only the location was present. For the third model, both dive method (R.I. = 0.25) and SCL (R.I. = 0.73) were present in the model. Notably, habitat type did not influence MAD. Upon evaluating parameters included in the top ranked model, the confidence interval for ln(SCL) just barely cross into the negative (Table 3).

For loggerhead sea turtles, the model confidence set included four top ranked models, and SCL and location were again the most influential factors (Table 4). For the top ranked model, the confidence interval for ln(SCL) was wholly positive. The confidence set models indicate that as the size of the sea turtle increases, the MAD increases. That is, larger turtles have a greater MAD and more often exhibit a startle response when researchers are on the upper range of MAD (i.e., >6 m), whereas the smaller individuals allowed the researchers to approach closer (within 2–4 m) before exhibiting a startle response. Dive method (snorkel vs. scuba) had a slight effect on the MAD, as it does appear in the confidence model set. However, the sample size of scuba surveys (*n* = 6) was small. With a larger sample size of more SCUBA surveys, the dive method may become a more influential factor for determining MAD.

## 4. Discussion

During our 16-month in-water survey, we successfully observed a range of behaviors among three sea turtle species, where size could be assigned to an individual. Body length linked to behavior is largely absent in previous in-water studies [18]. Thus, we were successfully able to provide novel insights on behaviors displayed at nearshore artificial reefs while discerning sea turtle wariness to researchers.

Our results suggest that larger individuals across species have a greater wariness response, becoming startled when the researcher was further away. In contrast, smaller individuals allowed researchers to get closer, approximately 1 m before eliciting a startle response, if a response was elicited at all. A major driver of antipredator behavior (i.e., wariness) across multiple taxa is the size of the individual, and this generally has a positive relationship with wariness [69]. Wariness has not been assessed in sea turtles prior to this study. However, wariness in fish using SVCs has been commonly studied and similar results have been found. For example, Andradi-Brown et al. (2018) found that a juvenile Acanthuridae (9.9 cm length) had a MAD of 2.05 m while a mature individual (23.2 cm long) had a MAD of 2.9 m, demonstrating a 0.85 m difference [70]. Studies on caimans and crocodiles have found similar results, in that wariness is affected by size of the individual [71,72,73]. Pacheco (1996) observed that as the size of caiman species (*Melonosuchus niger* and *C. yacare*) increased, wariness of individuals also increased; however, *M. niger* hatchlings showed little to no avoidance during spotlight studies, confirming a similar study in which *Alligator mississippiensis* hatchlings showed little to no avoidance [73]. Wariness is either a learned or instinctual behavior that can be influenced by different experiences as individuals in populations are subjected to natural or anthropogenic disturbance or risk (i.e., predator presence/interactions, diving pressures, hand-capturing efforts, or fishing pressures; [74]). Larger individuals may potentially be habituated to significant predation pressures leading to an increased perception of fear, especially when being approached by researchers/divers that appear to be larger than them [69]. Additionally, an increase in wariness in larger sea turtles could be an innate response to being a higher quality prey item with increased visibility and/or reduced escape speed when compared to smaller individuals [69].

Juvenile green sea turtles exhibited the most no startle responses and some of the shortest MADs (Figure 5). Some green sea turtles observed within the nGOM are likely residents, as they have been observed across sites (confirmed by photo-identification), presence of flipper tags, and have been admitted to rehabilitation facilities for fishing related injuries numerous times (T. Siegfried, pers. observation). At nearshore artificial reefs in the nGOM, green sea turtles were the most frequently resighted species, with an 8% resighting rate during SVC surveys from 2019 to 2022 (Roberto et al. *in preparation* [75]). Lamont and Johnson (2021) observed a similar trend while netting sea turtles over sandy bottom habitat in the nGOM; Greens had a re-capture rate of 30.8%, which was the highest of all the sea turtle species captured [74]. All our study sites are popular snorkel and scuba diving sites that frequently have relatively large numbers of divers daily, especially during tourist season in the summer. The increased interactions with divers and snorkelers could be a learned behavior, as resident juvenile green sea turtles become habituated to divers (Gulfarium CARE Center, Roberto et al., *in preparation* [75]). The loggerhead sea turtles we observed were composed mostly of subadult and adult size classes, which are inherently larger than the juvenile green sea turtles; therefore, this could explain their increased wariness during our study (Figure 5). However, additional studies comparing sea turtle behavior at sites frequently visited by humans versus sites that are more secluded would give better insight into how the presence or absence of humans affects sea turtle behavior, similar to past studies conducted for fish species [57,70,76].

Our SVC approach provides insight into sea turtle behavior and yields a more complex picture into sea turtle behavior than previously recorded. We observed behaviors rarely documented, including cleaning, feeding, and intraspecific interactions. Self-cleaning behavior has been documented for green sea turtles [31] and hawksbills [77], but observations of loggerhead sea turtles are rare. To our knowledge, only one other study, Schofield et al. (2014) [18], suggested that loggerhead sea turtles engaged in self-cleaning behaviors, as noted by scratch marks consistent with a rubbing pattern observed on their carapace. However, this is the first documented and recorded video of size-specific loggerhead sea turtles engaging in self-cleaning behaviors. This research contributes to a growing body of literature, using novel technology, such as SVCs and animal-borne cameras, which are yielding novel insights into the lives of sea turtles in their marine habitats, progress that has been a long-time coming for these threatened and challenging to study in situ, but charismatic animals.

## 5. Conclusions

The results of our study support the use of SVCs as a generally accessible, non-invasive tool to conduct ecologically relevant in-water surveys of sea turtles that can link behavioral observations to body size. We have provided a baseline dataset on sea turtle wariness at nearshore artificial habitats deployed in the nGOM. This approach can be used to better understand how behavior can change in response to disturbance or conservation intervention (i.e., new habitat addition, habitat restoration, establishment of a new marine reserve, etc.). Further investigation using SVCs on sea turtle behavior can help discern sublethal impacts on the sea turtle populations. Sublethal impacts can include diving pressure leading to turtle avoidance of reefs (perhaps leading to more frequently inhabiting fishing piers, introducing them to increased fishing pressures), or change in behavior (increased tolerance of divers over time). As new artificial reefs are continuously deployed, monitoring sea turtle presence and behavior over time as the reef becomes more frequented by divers and fishers would then allow for greater understanding of sea turtle behavior and wariness.

## Figures and Tables

**Figure 1 animals-13-00114-f001:**
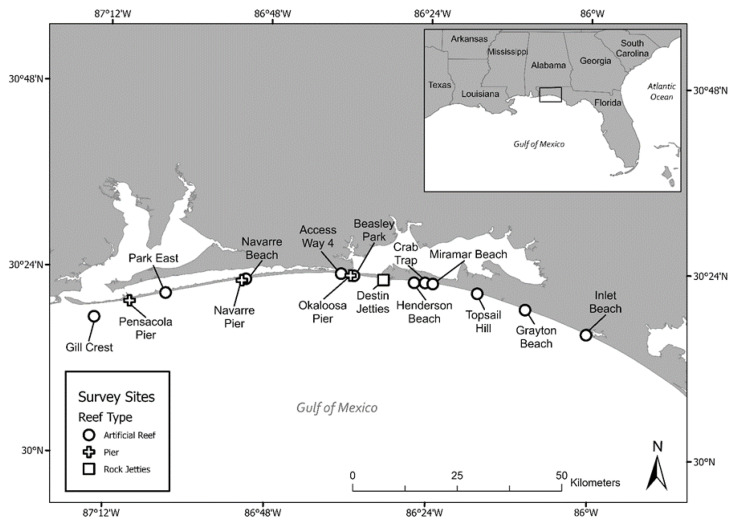
Map indicating field sites along the nGOM from Gill Crest Reef located off Pensacola Beach, Florida to Inlet Beach west of Panama City Beach, Florida.

**Figure 2 animals-13-00114-f002:**
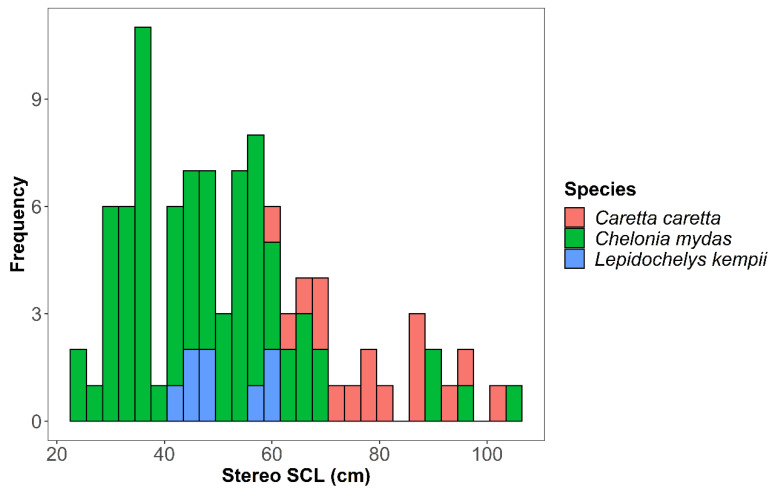
Size frequency distribution of three species of sea turtle observed across study sites in the northern Gulf of Mexico.

**Figure 3 animals-13-00114-f003:**
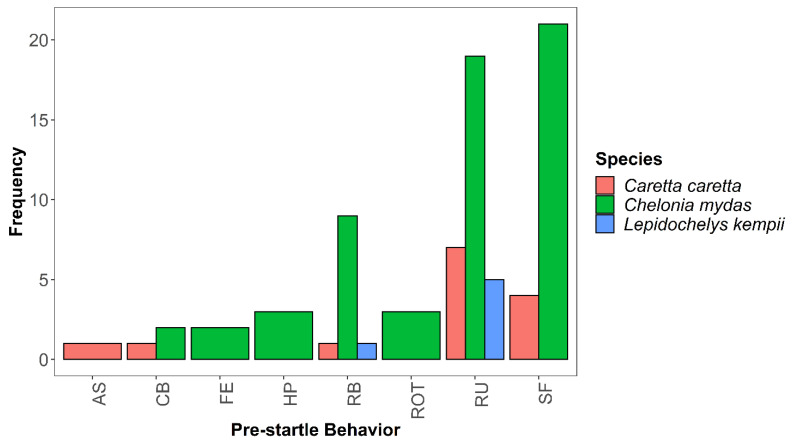
Frequency of pre-startle behavior across three sea turtle species (*Caretta caretta, Chelonia mydas,* and *Lepidochelys kempii*). Abbreviations: AS–ascent swimming, CB–crawling on benthos, FE–feeding, HP–holding position (no moving), RB–resting on benthos, ROT–resting on top of reef module/substrate, RU–resting under reef structure, SF–swimming forward. Turtles that did not exhibit a startle response are not represented here.

**Figure 4 animals-13-00114-f004:**
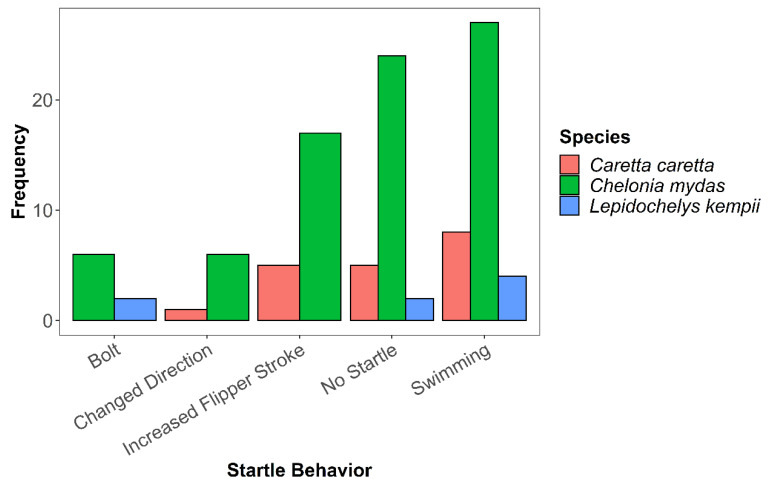
Frequency of startle behavior type across three species of sea turtles observed at study sites.

**Figure 5 animals-13-00114-f005:**
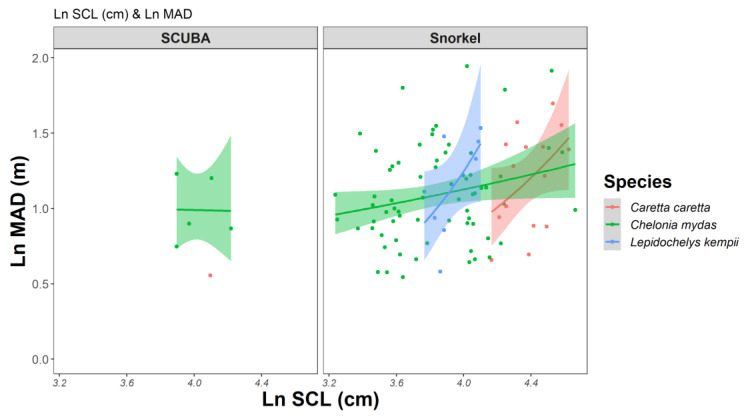
Linear mixed effects models (LM) of log-transformed MAD (m) and log transformed SCL (cm). The plots are differentiated by dive method and species of sea turtle observed. The solid line represents the predicted fit with shading for the 95% confidence interval.

**Table 1 animals-13-00114-t001:** Sea turtle behaviors and descriptions. Note: we include here reference videos providing examples of each behavior type (see Appendix A). Appendix A are provided in the description.

Behavior Category	Behavior Sub-Category	Description
Movement	Swimming forward	Movement through the water while maintaining the same depth in the water column (Appendix A)
	Diving	Steadily increasing depth (Appendix A)
	Ascent	Steadily decreasing depth (Appendix A)
	Crawling on benthos	Use of flippers against the seafloor to change position (Appendix A)
Obligatory	Breathing	Ascent in which the turtle’s head rises above the surface of the water (Appendix A)
	Feeding	Includes acts of foraging and visible consumption of organisms (Appendix A)
Unique	Intraspecific encounters	Interaction between sea turtles of the same species in which each is aware of the presence of the other (Appendix A)
	Self-cleaning	Active movement to scrape the surface of the sea turtle’s carapace or other body part on another surface (i.e., reef module) (Appendix A)
	Interspecific encounter with cleaner species	Symbiotic interaction between cleaner species (i.e., fish) and sea turtle to remove epibionts (Appendix A)
	Interspecific encounter with >1 species of sea turtle	Two species of sea turtle interacting with each other in proximity (Appendix A)
Resting	Resting underneath module	Not actively moving flippers and is remaining below the reef module (Appendix A)
	Resting on top of module	Not actively moving flippers and is remaining atop the reef module (Appendix A)

**Table 2 animals-13-00114-t002:** Minimum Approach Distance (MAD) ranges, mean ±SD, for all species of sea turtles. SD = standard deviation.

Species	Sample Size	MAD Range (m)	Mean ± SD
Green sea turtle	73	0.72–5.99	2.10 ± 1.10
Loggerhead sea turtle	16	0.93–3.80	2.12 ± 0.99
Kemp’s Ridley sea turtle	8	0.78–3.63	2.35 ± 0.99

**Table 3 animals-13-00114-t003:** Confidence set for linear mixed effects model examining the relationship of green sea turtle minimum approach distance and biological factors (ΔAICc < 2). Note: (+) symbol indicates variable was included in the model and (-) indicates variable was not included in the model. AICc = Akaike’s Information Criterion corrected for small sample size, ΔAICc = difference in AICc from the top ranked model, R.I. = relative importance.

Model Terms					Model Support			
*Chelonia*	Dive Method	Habitat Type	lnSCL (cm)	Location	df	AICc	ΔAICc	Weight
Model 1	-	-	0.19 (−0.01–0.44)	+	4	38.79	0	0.48
Model 2	-	-	-	+	3	39.93	1.14	0.27
Model 3	+	-	0.21	+	5	40.09	1.30	0.25
*R.I.*	0.25	-	0.73					

**Table 4 animals-13-00114-t004:** Confidence set for linear mixed effects model examining the relationship of loggerhead sea turtle minimum approach distance and biological factors (ΔAICc < 2). Note: (+) symbol indicates variable was included in the model and (-) indicates variable was not included in the model. AICc = Akaike’s Information Criterion corrected for small sample size, ΔAICc = difference in AICc from the top ranked model, R.I. = relative importance.

Model Terms					Model Support			
*Caretta*	Dive Method	Habitat Type	lnSCL (cm)	Location	df	AICc	ΔAICc	Weight
Model 1	-	-	1.24 (0.29–2.32)	+	4	20.37	0	0.38
Model 2	+	-	-	+	4	20.97	0.59	0.28
Model 3	-	-	-	+	3	21.68	1.31	0.20
Model 4	+	-	0.89	+	5	22.22	1.85	0.15
*R.I.*	0.43	-	0.53					

## Data Availability

The data presented in this study will be made publicly available on Zenodo at doi TBD, reference number TBD upon paper acceptance.

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
