# Peer review of "Size-Mediated Sea Turtle Behavioral Responses at Artificial Habitats in the Northern Gulf of Mexico"

_animals, 2022, doi:10.3390/ani13010114_

Round 1

Reviewer 1 Report

The paper "Using Minimum Approach Distance to Quantify size-mediated 2 Sea Turtle Response Behavior at artificial reefs in the northern 3 Gulf of Mexico" is a well conceived, well written study using state-of-the-art methodology and studying behavior of sea turtles in an ethically sound way, without  unnecessary disturbance or suffering of the animals. The behavioral studies tend to be time and resource consuming, and the authors managed to use the optimal amount of time to cover a large number of localities of different type. The figures are simple and easy to understand and the statistical methods are appropriate and well explained.  I didn't notice many errors in the manuscript itself, which means that the authors did a good job at carefully proofreading the manuscript prior to submission, but there are some moderate formatting issues in the References section. I have only a few questions that could be addressed by the authors in the Introduction or Discussion section, to clarify some things regarding the design of the study:

1. The authors deliberately choose to work on the localities that are anthropogenic in nature (like artificial reefs), or frequented by people - swimmers, divers, etc. They discuss that the animals may have been habituated to the human presence and it certainly affects the MAD. Are different results expected in the habitats with less anthropogenic presence? Are there examples of studies that compared MAD between the habitats with and without human presence? 

2. Do authors expect the similar ratio between the overall size and MAD in habitats with and without the human presence? Also, if there are available published examples, it would be fine to cite them.

More specific comments are in the annotated .pdf file.

Overall, I think that this paper is a solid contribution to the field and should be published in Animals after a minor revision.

Author Response

Response to Reviewer 1 Comments

Point 1: The authors deliberately choose to work on the localities that are anthropogenic in nature (like artificial reefs), or frequented by people - swimmers, divers, etc. They discuss that the animals may have been habituated to the human presence and it certainly affects the MAD. Are different results expected in the habitats with less anthropogenic presence? Are there examples of studies that compared MAD between the habitats with and without human presence? 

Response 1: While this is only hypothetical, as we do not have data from remote or uninhabited sites, we would hypothesize that different results are not expected in areas with few humans present. Ultimately, the main driver for the MAD is predator-prey avoidance, where predators could be other marine species or humans. Across many species, including in this study, the larger the individual, the more wary they become, especially when in the presence of a perceived predator. For example, in comparing fish MAD in areas that are fished and unfished areas, the results are the same. The larger fish have shorter MADs and this relationship between body length and MAD is often stronger in fished areas versus non-fished areas (see Gotanda et al., 2009; Andradi-brown et al., 2018, Goetz et al., 2017). Presumably, smaller individuals have not yet learned or experienced the perceived risk of predation when continuously under the protection of artificial habitats, unlike their larger migratory counterparts, who often travel miles in open ocean. However, we do not mention this in our study as we cannot truly compare without studying the impact or (near) pristine versus more urban sites, specifically. This could be another great study, using stereo-video cameras to assess areas that are highly populated versus areas that are not. We did add a sentence to lines 389-391 to emphasize the need for additional studies to address this question.

Point 2: Do authors expect the similar ratio between the overall size and MAD in habitats with and without the human presence? Also, if there are available published examples, it would be fine to cite them.

Response 2: We can only hypothesize that we would see similar ratios between overall size and MAD in habitats with and without human presence. Moreover, we hypothesis that we would see a stronger relationship between the MAD and size in areas with human presence than not, further supported that larger individuals have a shorter MAD, regardless of human presence.

Line comments:

Point 3: I agree, and it was sensible decision not to include it in the statistical analysis

Response 3: Thank you for your feedback.

Point 4: You cite R packages in two different ways here, it should be uniform. Lines 212,214

Response 4: Thank you for your feedback. We have corrected as suggested

Point 5: Is this elaborated in the discussion? And how could location (but not habitat type) affect? Line 298

Response 5: Thank you for your feedback. We have clarified statements regarding the rationale for how we set up the statistical model in the methods. We clarified statements on how the results (with habitat type not presented in the models in the confidence set) in the results. We believe that this goes in deeper to the statistically analysis reasoning than our paper addressed. We notated that location was used as a repeated effect because of multiple visits to eat site. Treating it as a repeated effect allows us to account for potential spatial autocorrelation as we repeatedly visited sites over time during the survey period. However, using the site as a repeated effect could serve as a proxy for other kinds of variables we did not measure in our study, e.g. water temperature, depth, water current flow, food availability, etc. Locations are more complex than just the habitat classification that we give them. On the other hand, habitat type classifies the different types of habitat we visited, artificial reefs, fishing piers, and jetties, but, as this variable was not present in the confidence set, it is possible that the different kinds of artificial structures don’t influence MAD for sea turtles. In other words, their wariness response to divers approaching them is independent of habitat type.

Point 6: Acanthuridae? Line 339

Response 6: Thank you for your feedback. We have corrected as suggested.

Point 7: Italicize Caretta caretta. Line 471

Response 7: Thank you for your feedback. We have corrected as suggested.

Point 8: Italicize Chelonia mydas Line 474

Response 8: Thank you for your feedback. We have corrected as suggested.

Point 9: Format this reference the same way as the other ones.   Line 509

Response 9: Thank you for your feedback. We have corrected as suggested.

Point 10: The journal names are abbreviated in some cases and written out full in other cases. Please be consistent.  Line 532

Response 10: Thank you for your feedback. We have corrected as suggested

Point 11: Again, the journal name is abbreviated in the previous reference and given in full here. Line 557

Response 11: Thank you for your feedback. We have corrected as suggested

Point 12: Italicize Line 563

Response 12: Thank you for your feedback. We have corrected as suggested.

Point 13: Pagination? Line 577

Response 13: Thank you for your feedback. We have corrected as suggested.

Reviewer 2 Report

General comments

In this manuscript, the authors use stereo-video cameras to record behavioural patterns in sea turtles, and link these to morphometric data to assess changes in behavioural responses across various life stages. As a case study, the minimum approach distance of observers/divers were linked to body size, using three syntopic turtle species residing in the Gulf of Mexico as a study system. Besides adding evidence towards a proof-of-concept for using stereo-video cameras as a viable and non-invasive method in sea turtle behavioural studies, its recorded avoidance behaviour might furthermore have relevance for conservation efforts in (artificial) reefs, as it addresses the potential implications of increased recreational pressure.

Overall, the manuscript is concise and well-written. Its methods and the described modelling approach are relatively straightforward and generally well-executed, and I think only minor corrections are necessary before it meets the standard of publication.

Specific comments

Title: Instead of a more technical title, it could be elaborated/edited to better reflect the central use of stereo-cameras as a non-invasive method, although this is of course up to the discretion of the authors. The capitalization of words is furthermore not consistent yet in the version provided.

L 44-46: The keywords have some numbers in between that still need to be removed.

L 49: To avoid confusion with only the development until adulthood/reproductive age, ‘ontogeny’ could be replaced with ‘lifespan’ or a similar term.

L 62: A space is missing in before the references.

L 68: Would break this up a bit for general readability: ‘video-based in-water and non-invasive’.

L 69-71: I would suggest adding some brief information about SVCs for more unfamiliar readers in the introduction or methods, especially since they present the central data recording equipment in this study; e.g. highlight that they render a 3-dimensional image, which is essential in understanding they allow an estimation of body size.

L 82-83: I would suggest using something like ‘challenging’ as a synonym for ‘depauperate’ in this sentence, which reads as a bit awkward in this context.

L 83: ‘analyses’?

L 87-89: This sentence rather repeats the one before. It might be better to state a few examples where turtles have positive trophic effects on an ecosystem scale.

L 115-116: Not sure if I understand this correctly; perhaps something like “with dive sites selected opportunistically and visited on a rotating basis”? Please briefly clarify, e.g. if an attempt was made to visit all sites a similar amount of times.

L 181-186: It does not completely become clear to me how the quantification of startling intensity is used in further analyses, and whether the medium category (1) was treated differently from 0 or 2.

Table 3: This table is not very readable at the moment, which is more a formatting issue to be addressed by the copyediting team - better be made as wide as tables 1+2.

L 374-375: This is indeed a very noteworthy point of the study. However, these findings are a bit more of an ‘add-on’, as these more specific behaviours were – to my knowledge – not included in subsequent analyses and the modelling pipeline (besides those indicating startling behaviours). I would therefore suggest highlighting these more clearly as noteworthy natural history observations in abstract or the results, which add to the use of stereo cameras in recording such information, but were not included in formal analyses.

L 529-533: Not sure whether this reference format is correct. Might be more suitable to cite this following more specific guidelines for a book chapter?

Author Response

Response to Reviewer 2 Comments

Point 1: Instead of a more technical title, it could be elaborated/edited to better reflect the central use of stereo-cameras as a non-invasive method, although this is of course up to the discretion of the authors. The capitalization of words is furthermore not consistent yet in the version provided.

Response 1: Thank you for your feedback. We have revisited our title and changed it to be less technical.

Point 2: Keywords have some number in between that still need to be removed L44-46

Response 2: Thank you for your feedback. We have corrected as suggested.

Point 3: Replace ontogeny with lifespan L49

Response 3: Thank you for your feedback. We have corrected as suggested.

Point 4: A space is missing in before the references L62

Response 4: Thank you for your feedback. We have corrected as suggested.

Point 5: I would suggest adding some brief information about SVCs for more unfamiliar readers in the introduction or methods, especially since they present the central data recording equipment in this study; e.g. highlight that they render a 3-dimensional image, which is essential in understanding they allow an estimation of body size. L69-71

Response 5: Thank you for your feedback. We have corrected as suggested. Please see lines 68   – 70.

Point 6: Change despauperate to challenging L82-83

Response 6: Thank you for your feedback. We have corrected as suggested.

Point 7: Change analysis to analyses L83

Response 7: Thank you for your feedback. We have corrected as suggested.

Point 8: This sentence repeats the one before it. It might be better to state examples where turtles have positive trophic effects on an ecosystem scale. L87-89

Response 8: Thank you for your feedback. We have removed the beginning sentence to eliminate the repetitiveness.

Point 9: Not sure if I understand this correctly; perhaps something like “with dive sites selected opportunistically and visited on a rotating basis”? Please briefly clarify, e.g. if an attempt was made to visit all sites a similar amount of times. L115-116

Response 9: Thank you for your feedback. We have added to lines 114 stating that we tried to visit all sites equally.

Point 10: It does not completely become clear to me how the quantification of startling intensity is used in further analyses, and whether the medium category (1) was treated differently from 0 or 2. L181-186

Response 10: Thank you for your feedback. In further analysis, startle intensities 1 and 2 were not treated any differently. Ultimately, we used a binary response (startle or no startle) as an explanatory variable to predict the MAD. We have updated in the methods to reflect this. Line: 182-183, 189-190.

Point 11: Table 3 not readable

Response 11: Thank you for your feedback. We have tried to format more correctly to assist in the copyediting team.

Point 12: This is indeed a very noteworthy point of the study. However, these findings are a bit more of an ‘add-on’, as these more specific behaviours were – to my knowledge – not included in subsequent analyses and the modelling pipeline (besides those indicating startling behaviours). I would therefore suggest highlighting these more clearly as noteworthy natural history observations in abstract or the results, which add to the use of stereo cameras in recording such information, but were not included in formal analyses. L374-375

Response 12: Thank you for your feedback. We have added a sentence into the results indicated that we did not use these behaviors in the analysis see lines 280-281 "While this may have been the first recorded instance of size-specific behaviors observed these rarely recorded behaviors were not included in the formal statistical analysis.

Point 13: Reference formatting L529-533

Response 13: Thank you for your feedback, we agree and have updated the reference more specific for a book chapter.